# OX2R-selective orexin agonism is sufficient to ameliorate cataplexy and sleep/wake fragmentation without inducing drug-seeking behavior in mouse model of narcolepsy

Hikari Yamamoto[1], Yasuyuki Nagumo[1], Yukiko Ishikawa[1], Yoko Irukayama-Tomobe[1], Yukiko Namekawa[1], Tsuyoshi Nemoto[2], Hiromu Tanaka[2], Genki Takahashi[2], Akihisa Tokuda[1], Tsuyoshi Saitoh[1], Hiroshi Nagase[1], Hiromasa Funato[1,3], Masashi Yanagisawa[1,4,5,6]*

1 International Institute for Integrative Sleep Medicine (WPI-IIIS), University of Tsukuba, Tsukuba, Ibaraki, Japan, 2 School of Medicine, University of Tsukuba, Tsukuba, Ibaraki, Japan, 3 Department of Anatomy, Faculty of Medicine, Toho University, Ota-ku, Tokyo, Japan, 4 Department of Molecular Genetics, University of Texas Southwestern Medical Center, Dallas, Texas, 5 Life Science Center, Tsukuba Advanced Research Alliance, University of Tsukuba, Tsukuba, Ibaraki, Japan, 6 R&D Center for Frontiers of MIRAI in Policy and Technology, University of Tsukuba, Tsukuba, Ibaraki, Japan

* yanagisawa.masa.fu@u.tsukuba.ac.jp

## Abstract

Acquired loss of hypothalamic orexin (hypocretin)-producing neurons causes the chronic sleep disorder narcolepsy-cataplexy. Orexin replacement therapy using orexin receptor agonists is expected as a mechanistic treatment for narcolepsy. Orexins act on two receptor subtypes, OX1R and OX2R, the latter being more strongly implicated in sleep/wake regulation. However, it has been unclear whether the activation of only OX2R, or both OX1R and OX2R, is required to replace the endogenous orexin functions in the brain. In the present study, we examined whether the selective activation of OX2R is sufficient to rescue the phenotype of cataplexy and sleep/wake fragmentation in orexin knockout mice. Intracerebroventricular [Ala$^{11}$, $_D$-Leu$^{15}$]-orexin-B, a peptidic OX2R-selective agonist, selectively activated OX2R-expressing histaminergic neurons in vivo, whereas intracerebroventricular orexin-A, an OX1R/OX2R non-selective agonist, additionally activated OX1R-positive noradrenergic neurons in vivo. Administration of [Ala$^{11}$, $_D$-Leu$^{15}$]-orexin-B extended wake time, reduced state transition frequency between wake and NREM sleep, and reduced the number of cataplexy-like episodes, to the same degree as compared with orexin-A. Furthermore, intracerebroventricular orexin-A but not [Ala$^{11}$, $_D$-Leu$^{15}$]-orexin-B induced drug-seeking behaviors in a dose-dependent manner in wild-type mice, suggesting that OX2R-selective agonism has a lower propensity for reinforcing/drug-seeking effects. Collectively, these findings provide a proof-of-concept for safer mechanistic treatment of narcolepsy-cataplexy through OX2R-selective agonism.

**Data Availability Statement:** All relevant data are available within the paper and its Supporting Information files.

**Funding:** This study was funded by a grants from the World Premier International Research Center Initiative from Ministry of Education, Culture, Sports, Science and Technology, the Japan Society for the Promotion of Science (grant number 17H06095), the Core Research for Evolutional Science and Technology (grant number A3A28043), theFunding Program for World-Leading Innovative R&D on Science and Technology (FIRST Program) from JSPS, the Uehara Memorial Foundation, and the Japan Agency for Medical Research and Development (grant number JP21zf0127005), all awarded to MY. The funders had no role in study design, data collection and analysis, decision to publish, or preparation of the manuscript.

**Competing interests:** The authors have declared that no competing interests exist.

# Introduction

Narcolepsy type-1 (also known as narcolepsy-cataplexy) can be interpreted through two distinct pathophysiological phenomena, rapid eye movement (REM) sleep-related and non-rapid eye movement (NREM) sleep-related symptoms. The abnormal gating of REM sleep-related neurophysiological mechanisms, such as REM atonia, contributes to cataplexy (sudden bilateral skeletal muscle weakening triggered by a strong emotion) and sleep paralysis. On the other hand, the abnormal regulation of NREM sleep onset results in sleep/wake fragmentation characterized by frequent transitions between wakefulness and NREM sleep [1, 2]. Clinically, the latter is manifested as excessive daytime sleepiness often with "sleep attacks," and frequent nocturnal awakening. Narcolepsy-cataplexy is currently treated with symptomatic pharmacotherapy using psychostimulants (e.g., modafinil and methylphenidate), sedatives (e.g., sodium oxybate), antidepressants (e.g. venlafaxine and clomipramine), and histarminergics (e.g., pitolisant), and thus there is no fundamental mechanistic treatment available [3, 4]. The use of these medications is often limited by adverse effects such as headache, nausea, palpitations, anxiety, irritability and insomnia. Therefore, novel medications for narcolepsy are highly desired.

Patients with type-1 narcolepsy exhibit autoimmune loss of orexin (hypocretin) neurons in the lateral hypothalamus (LH) [5, 6]. Orexin levels in patients' cerebrospinal fluid are low and often undetectable [6, 7]. Narcoleptic symptoms in orexin neuron-ablated mice can be ameliorated by intracerebroventricular (ICV) administration of orexin peptides [8]. These findings suggest that the pharmacological activation of orexin receptors is of potential value for treating human narcolepsy. Orexin peptides, orexin-A (OXA) and orexin-B, act on two types of G protein-coupled receptors, the orexin receptors type-1 (OX1R) and type-2 (OX2R) [9]. Importantly, OX2R-mediated pathway is crucial for stability of wakefulness, whereas OX1R has only marginal roles on sleep/wake regulation when OX2R signaling is intact [10, 11]. Mice lacking OX1R show essentially normal sleep/wake phenotype whereas mice lacking both OK1R and OX2R show severe sleep/wake fragmentation and direct wake-to-REM transitions with cataplexy-like episodes [12, 13]. On the other hand, OX1R has other complex physiological roles such as emotional and motivational regulation. OX1R has also been implicated in the dopaminergic reward pathway, addictive behaviors, and fear memory [14–18].

We previously reported that a nonpeptidic OX2R-selective agonist, YNT-185, promotes wakefulness and ameliorates cataplexy-like episodes in mouse models of narcolepsy such as orexin knockout and orexin neuron-ablated mice [19]. However, YNT-185 did not sufficiently suppress the active-phase fragmentation of wakefulness (another symptom of narcolepsy) in the narcoleptic model mice. Interestingly, a previous report indicates that the sleep/wake fragmentation in orexin receptor dual knockout mice is rescued by restoration of OX1R in the locus coeruleus (LC) [20]. Thus, it is possible that not only OX2R but also OX1R may play a crucial role in preventing sleep/wake fragmentation. It remains to be determined whether the selective activation of OX2R is sufficient to prevent narcoleptic symptoms including cataplexy and sleep/wake fragmentation, or additional activation of OX1R is also required in order fully to ameliorate these symptoms.

In the present study, to examine and verify the fundamental strategy for treating narcoleptic symptoms including cataplexy and sleep/wake fragmentation, we compared the therapeutic efficacies of peptidic OX2R-selective and OX1R/OX2R non-selective agonists side-by-side in narcoleptic orexin knockout mice. We demonstrated that ICV injection of [Ala$^{11}$, $_D$-Leu$^{15}$]-orexin-B (AL-OXB), a selective and potent OX2R agonist [21], suppresses fragmentation of wakefulness as well as cataplexy-like episodes without in-vivo OX1R activation in orexin knockout (OXKO) mice. In addition, we showed that AL-OXB does not induce addiction-like

drug-seeking behaviors as compared to OXA, a non-selective OX1R/OX2R agonist. Our data suggest that the selective activation of OX2R is sufficient to ameliorate major narcoleptic symptoms. Furthermore, OX2R agonism may be superior to OX1R/OX2R dual agonism in terms of safety, because of the apparently lower risk of psychological/addictive side effects.

## Material and methods

### Animals

All the animal experiments were approved by the Animal Experiment and Use Committee of the University of Tsukuba (Protocol Number: 21–313). All surgery was performed under iso-flurane anesthesia, and all efforts were made to minimize suffering. We used 8–16 week-old male $Hcrt^{-/-}$ mice with a C57BL/6J genetic background (abbreviated as OXKO) for all sleep recording experiments (a total of 50 mice) [22]. We also used 7-week-old male wild-type C57BL/6J mice for the conditioned place preference test (a total of 48 mice). All mice were kept on a 12-h light/dark cycle (lights on at 9:00 A.M.) at an ambient temperature of $23.5 \pm 2.0°C$ under specific pathogen-free conditions.

### Reagent

[Ala$^{11}$, $_D$-Leu$^{15}$]-human orexin-B (AL-OXB, >98% pure) [21] and human orexin-A (OXA, >99% pure) (Peptide Institute Inc., Osaka, Japan) were dissolved in saline. In $Ca^{2+}$ mobiliza-tion assay, these peptides were dissolved in 0.1% BSA (Sigma-Aldrich) / phosphate buffered saline (PBS) as 100-μM stock solutions. The administered peptide amounts are indicated in terms of net peptide content.

### Intracellular $Ca^{2+}$ mobilization assay

$Ca^{2+}$ mobilization assay was conducted as described previously [19]. Chinese hamster ovary-K1 (CHO-K1) cells stably expressing human OX1R or OX2R were seeded in a 96-well plate (10,000 cells per well) and incubated with Dulbecco's modified Eagle's medium (WAKO) con-taining 5% fetal bovine serum (Corning), 1% penicillin/streptomycin (WAKO), 1×NEAA (Gibco, 11140050), 1mM G418 (WAKO) and 0.8μM Puromycin (Sigma-Aldrich) at 37°C for 48 h. Then, cells were loaded with 5 μM fluorescent calcium indicator Fura 2-AM (Cayman Chemical) in HBSS (GIBCO) including 20 mM HEPES (Sigma-Aldrich), 2.5 mM Probenecid (Sigma-Aldrich), 0.04% CremophorEL (Fluka), and 0.1% BSA at 37°C for 1 h. Cells were washed and 75 μL of HBSS buffer was added. Then, cells were treated with 25 μL of AL-OXB or OXA in the Functional Drug Screening System 7000 (Hamamatsu Photonics). Increase in the intracellular $Ca^{2+}$ concentration was measured from the ratio of fluorescence emission at 510 nm by excitation at 340 or 380 nm. EC$_{50}$ values were calculated from dose-response curves by using GraphPad Prism 7.

### Implantation of EEG/EMG electrode and ICV guide cannula

Male narcolepsy model mice (8 weeks) were implanted with an EEG/EMG electrode contain-ing two stainless steel screws and two flexible wires as described previously [19], under anes-thesia using isoflurane (4% for induction, 2% for maintenance). Two stainless steel screws (1.0 mm diameter) were used for EEG electrodes, one of which was placed epidurally over the right frontal cortex (5.03 mm anterior and 1.27 mm lateral to lambda) and the other over the right parietal cortex (1.27 mm lateral to lambda) under stereotaxic control. Two flexible wires were implanted into both trapezius muscles (left/right) for EMG recording. For ICV administration, mice were simultaneously implanted with a guide cannula into left lateral ventricle (0.3 mm

posterior and 0.9 mm lateral to bregma, 2.2 mm depth from skull surface) as described [8, 19]. The whole assembly was then attached to the skull with dental cement. After recovery from anesthesia, the mice were housed individually, let to recover for at least 1 week and then connected to a tether hung from a counterbalanced arm (Instech Laboratories) that allowed the free movement for habituation to the recording conditions for another 1 week.

### ICV administration

ICV administration with EEG/EMG recordings was conducted as previously described [8, 19]. Single ICV administration for EEG/EMG recordings was performed under anesthesia using isoflurane (4% for induction, 2% for maintenance) at zeitgeber time (ZT) 11–12. The mice were administered through a guide cannula connected to an oil-filled microsyringe (Hamilton) by using an automated syringe pump (Harvard Apparatus). Vehicle and peptide solutions were injected into the left lateral ventricle in 3 μL over 6 min (flow rate: 0.5 μL/min) and the EEG/EMG signals were measured as mentioned below. Mice with single ICV administrations with EEG/EMG recording were given at intervals of at least 3 days to adopt a randomized cross-over design. Continuous ICV administration for EEG/EMG recording was performed without anesthesia from ZT12-24. The mice were maintained in administration for 12 h through a guide cannula connected to an oil-filled microsyringe (Hamilton) by using an automated syringe pump (Harvard Apparatus), which remained in place through the dark phase. Vehicle and peptide solutions were injected into the left lateral ventricle in 18 μL in 12 h (flow rate: 0.025 μL/min) while measuring EEG/EMG. Bolus ICV administration for conditioned place preference test was performed without anesthesia during light phase (ZT4-7). 2 days before the injection, mice were briefly anesthetized with isoflurane (4% for induction), and then a hole was made in the skull at 2.0 mm lateral and 2.0 mm posterior to bregma, 2.0 mm depth from skull surface. The coordinates were optimized for 7-week-old young adult mice; we had confirmed that blue ink was administered correctly to the lateral ventricle. On the day of the conditioning session in conditioned place preference, vehicle or peptide solutions was injected into the left lateral ventricle through the hole using the 2.0 mm double needle (Natsume Seisakusho Co. Ltd., Tokyo, Japan) attached to a 25 μl microsyringe (Hamilton) in 4 μl in a few seconds.

### EEG/EMG analysis

EEG/EMG data were visualized and semiautomatically analyzed by MATLAB-based software as previously described [19]. The vigilance state in each 20-s epoch was classified as wakefulness, NREM sleep or REM sleep. Wakefulness was scored based on the presence of fast EEG activity, high amplitude and variable EMG activity. NREM sleep was staged based on high amplitude, delta (1–4 Hz) frequency EEG wave and low EMG tones. REM sleep was characterized by theta (6–9 Hz)-dominant EEG oscillations and EMG atonia. Epochs containing two different vigilance states within 20-s epoch were given the score of the predominant state. EEG signals were subjected to a fast Fourier transform analysis from 1- to 30-Hz with a 1-Hz bin by using MATLAB-based custom software. Total time spent in wakefulness, NREM and REM sleep were derived by summing the total number of 20-s epochs in each stage. Mean episode durations were determined by dividing the total time spent in each stage by the number of episodes of that state. In the present study, a relatively long epoch length (20 s) for mice was used for sleep staging. When different sleep/wake states are co-present within an epoch, the epoch is scored to the predominant state—the state that occupy the longest period of time in the epoch. Scoring with 20-s epochs results in larger absolute values of mean sleep-stage episode durations (Figs 2F, 4D; S1B Fig) and smaller absolute numbers of stage transitions (Fig 3).

However, we previously verified that the use of 20-s epoch length is still valid for mutually comparing the sleep/wake characteristics among the mice subjected to various pharmacologic or genetic perturbations, yielding conclusions identical to those obtained with shorter epoch lengths such as 4 s [23, 24]. In the present study, we confirmed this point by rescoring in 4-s epochs the EEG/EMG recordings for Fig 2E and 2F. As shown in S1E and S1F Fig, analyses with 20-s and 4-s epochs yielded the same conclusions. Cataplexy-like state was identified based on the criteria as described [25]: (i) an abrupt episode of nuchal atonia lasting at least 10 s; (ii) theta activity dominates the EEG during the episode; (iii) at least 40 s of wakefulness precedes the episode. For evaluation of cataplexy-like state, OXKO mice were given about 3 g of milk chocolates (Hershey) just after ICV administration to trigger cataplexy as described [26]. In contrast, mice were not given milk chocolates in the evaluation of wake fragmentation.

## Immunohistochemistry

OXKO mice were injected saline or peptide solutions (3 nmol) at ZT11.5 by the same protocol as the single ICV administration with guide cannula for EEG/EMG analysis. At 1.5 h after ICV administration, mice were anesthetized with pentobarbital sodium (50 mg/kg, IP) and transcardially perfused with PBS containing 10% sucrose followed by 4% paraformaldehyde (PFA) dissolved in PBS. Brains were quickly removed and post-fixed with 4% PFA at 4°C overnight and immersed in 30% sucrose dissolved in PBS for 2 days. Brains were frozen in OCT Compound (Sakura Finetek, Torrance, CA) at -80°C. The frozen 80-μm coronal sections were serially cut with a cryostat (Leica, Germany). Brain sections were washed with PBS and incubated with 1% Triton in PBS at room temperature for 3 h. Then, the sections were blocked with 10% Blocking One (Nacalai Tesque) in PBS with 0.3% Triton X-100 (blocking solution) at room temperature for 1 h. The sections were incubated at 4°C overnight with rabbit anti-c-fos antibody (1:2000; Millipore, RRID: AB_2631318), mouse anti-tyrosine hydroxylase (TH) antibody (1:1000; Millipore, RRID: AB_2201528), guinea-pig anti-c-fos antibody (1:1000; Synaptic Systems, RRID: AB_2619946) and rabbit anti-histidine decarboxylase (HDC) antibody (1:1000; ORIGENE, RRID: AB_1002154) in PBS containing 10% Blocking One and 0.3% Triton X-100. After washing in PBS 3 times, sections were reacted with secondary antibodies to Alexa Fluor[TM]488 (for c-fos: donkey anti-rabbit IgG 1:1000, Invitrogen, RRID: AB_2535792; goat anti-guinea pig IgG antibody 1:1000, Invitrogen, RRID: AB_2534117) and Alexa Fluor[TM]594 (for TH: donkey anti-mouse IgG 1:1000, Invitrogen, RRID: AB_141633; for HDC: donkey anti-rabbit IgG 1: 1000, Invitrogen, RRID: AB_141637) at 4°C overnight. The sections were counter-stained with Neuro Trace fluorescent Nissl stain (1:500, Invitrogen, N-21479), and were washed 3 times in PBS, mounted, air dried and covered. All images were taken on a Zeiss LSM800 confocal microscope (Zeiss, Germany) with a 20X (NA = 0.8) objective lens using the Zen software (Zeiss). Brain regions were defined according to Franklin and Paxinos's atlas of the mouse brain [27]. A single optical Z section (18-μm thick) per slice near the top surface was photographed and used for quantitative analysis. The cells were counted manually by three investigators blinded to the assignment of the treatment group. Cell counts from all slices containing the LC or tuberomammillary nucleus (TMN) were accumulated in each brain. The percentages of c-fos positive neurons among all noradrenergic or histaminergic neurons counted in each brain were then calculated. For each treatment group, 4–5 mice were used.

## Conditioned place preference test

Conditioned place preference (CPP) test was conducted as described [28]. The test was conducted using a shuttle box (15 cm wide × 30 cm long × 15 cm high; Natsume Seisakusyo Co. Ltd.) that was made of acrylic resin board and divided into two equal-sized compartments

with different flooring and walls separated by removable board. One compartment was white with a textured floor, and the other was black with a smooth floor to create equally preferable compartments. The place conditioning schedule consisted of three phases (pre-conditioning test, conditioning, and post-conditioning test). During the pre-conditioning test, the partition separating the two compartments was raised to 7 cm above the floor, a neutral platform was inserted along the seam separating the compartments, and mice were placed on the platform and allowed to freely explore the full extent of the shuttle box for 900 s. The time spent in each compartment was then recorded automatically with an infrared beam sensor (KN-80; Natsume Seisakusyo Co. Ltd.). Conditioning sessions (3 days for orexins, 3 days for saline) were conducted once daily for 6 days. Immediately after ICV injection of saline, mice were released into the compartment in which they had spent the most time in the pre-conditioning test for 1 h. Next day, these mice were injected orexins and were placed in the other compartment for 1 h. On the day after the final conditioning session, the post-conditioning test was performed by the same method as the pre-conditioning test. CPP score was calculated by post-conditioning test score minus pre-conditioning score in the drug-conditioning side.

## Statistics

Statistical analysis was performed using Graph Pad Prism 7.0 (Graph Pad Software). All data were expressed as mean ± SEM. Normality of data distribution was confirmed with Shapiro-Wilk test. Statistical significance was determined by one-way ANOVA followed by Bonferroni's multiple comparisons test, or two-way repeated-measures ANOVA followed by Bonferroni's test. Probability (P) values less than 0.05 ($P < 0.05$) were considered statistically significant.

## Results

### AL-OXB selectively and potently activates OX2R in vitro and in vivo

AL-OXB, which is a modified orexin-B, was developed as a peptide with much higher selectivity for OX2R [21, 29]. To confirm the selectivity and efficacy of AL-OXB for OX2R, we first conducted intracellular $Ca^{2+}$ transient assays on CHO cells stably expressing human OX1R or OX2R. OXA dose-dependently induced intracellular $Ca^{2+}$ mobilization with comparable $EC_{50}$ values for OX2R and OX1R: the $EC_{50}$ values were 0.50 nM for OX1R and 0.20 nM for OX2R, respectively (Fig 1A). Under the same condition, AL-OXB selectively induced intracellular $Ca^{2+}$ mobilization in cells expressing OX2R in a dose-dependent manner: the $EC_{50}$ values were 58 nM for OX1R (Emax value = 99.54%) and 0.055 nM for OX2R (Emax value = 92.83%), respectively (Fig 1A). These results indicate that AL-OXB is a selective OX2R agonist with approximately 1000-fold selectivity for OX2R over OX1R.

We next examined whether AL-OXB selectively activates brain OX2R in narcoleptic pre-pro-orexin knockout (Hcrt[-/-], abbreviated as OXKO) mice in vivo [22]. Sleep and wakefulness are controlled by complex neural circuits [30], but here we focused on the activation of noradrenergic neurons in the LC and histaminergic neurons in the TMN as direct downstream targets of orexin neurons, considering that LC-noradrenergic neurons and TMN-histaminergic neurons exclusively express OX1R and OX2R, respectively [31]. We used c-fos immunoreactivity as a marker for neuronal activation. ICV injection of OXA significantly increased the proportion of c-fos-positive cells in TH-positive LC-noradrenergic neurons as well as in HDC-positive TMN-histaminergic neurons (Fig 1B and 1C) of OXKO mice. In contrast, AL-OXB significantly increased the proportion of c-fos-positive cells in HDC-positive TMN-histaminergic neurons, but not in TH-positive LC-noradrenergic neurons (Fig 1B and 1C). These data indicate that AL-OXB selectively promotes neuronal activity through OX2R in

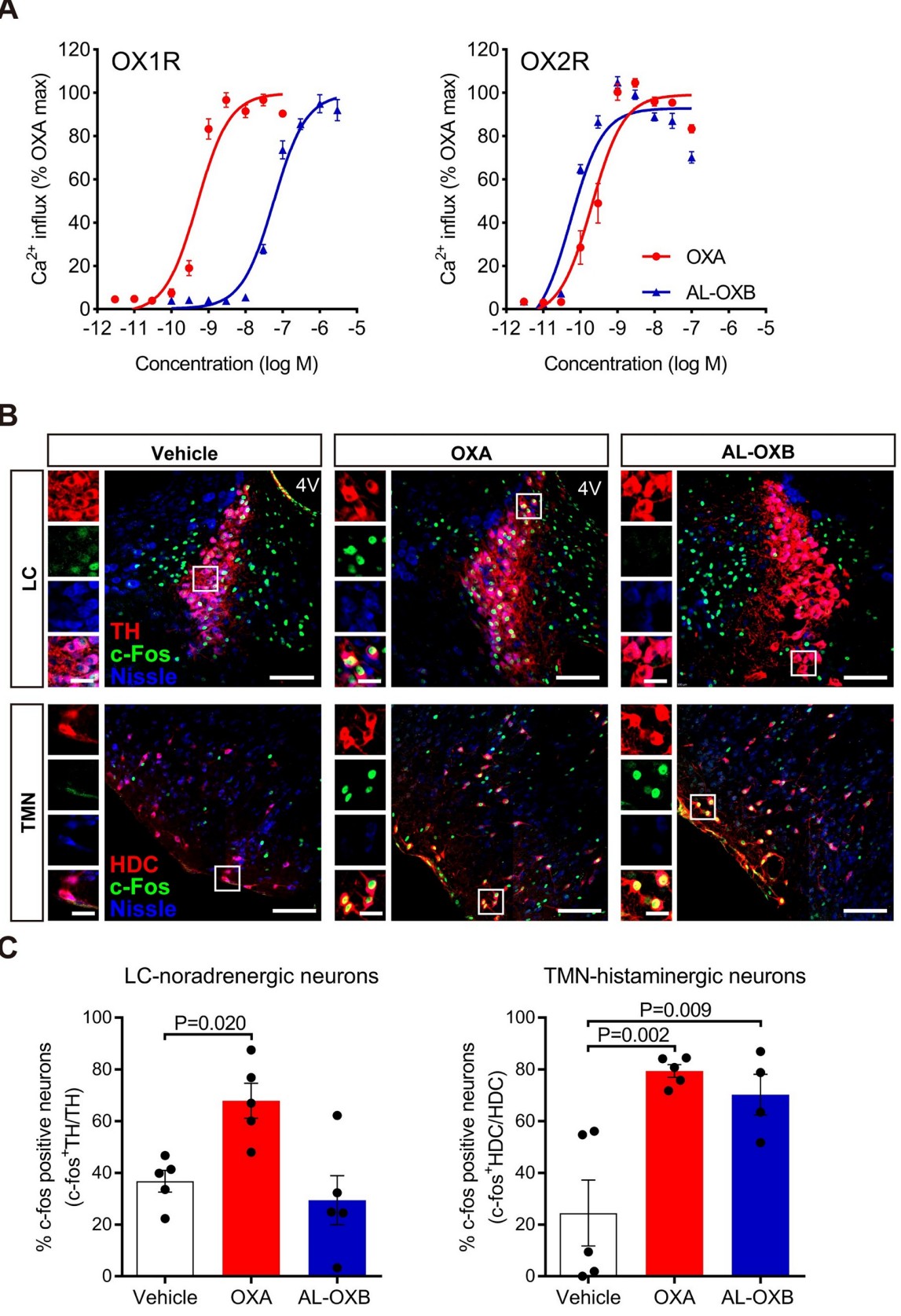

**Fig 1. [Ala[11], D-Leu[15]]-orexin-B (AL-OXB) selectively activates OX2R in vitro and in vivo. (A)** Dose-response curves of intracellular Ca$^{2+}$ transients induced by OXA and AL-OXB in CHO/OX1R cells (left) and CHO/OX2R cells (right). Data represent the mean ± SEM from two independent assays. **(B)** Representative images of c-fos immunoreactivity (green) after ICV administration of OXA or AL-OXB in tyrosine hydroxylase-positive (red) noradrenalin neurons of the LC and histidine decarboxylase-positive (red) histamine neurons of the TMN. 4V: 4th ventricle. Regions marked by white rectangles are magnified in small panels. Scale bars: small panels, 25 μm; large panels, 100 μm. **(C)** Quantification of c-fos-positive cells in LC-noradrenergic neurons (left) and TMN-histaminergic neurons (right). Data represent the means ± SEM from 4–5 mice. Statistical analysis: one-way ANOVA followed by Bonferroni's multiple comparisons test.

vivo. We thus confirmed that AL-OXB and OXA are selective OX2R agonist and non-selective OX1R/OX2R agonist in vivo, respectively.

## Selective activation of OX2R is sufficient to ameliorate narcoleptic symptoms in OXKO mice

Orexin-deficient narcoleptic humans and animals suffer from two major symptoms, cataplexy and sleepiness, in the active phase. In narcoleptic OXKO mice, these symptoms correspond to cataplexy-like states (see Methods) and sleep/wake fragmentation during the dark phase. We examined whether selective activation of OX2R by AL-OXB is sufficient to ameliorate cataplexy-like states in OXKO mice after single ICV administration within 1 h before the onset of the dark phase (ZT 11–12) (Fig 2A). Chocolate was given to OXKO mice in order to induce more frequent cataplexy-like states [26]. ICV administered OXA (3 nmol) significantly decreased the number of cataplexy-like states during 3 h after the injection and increased the latency to the first cataplexy-like state. Under the same condition, ICV administered AL-OXB

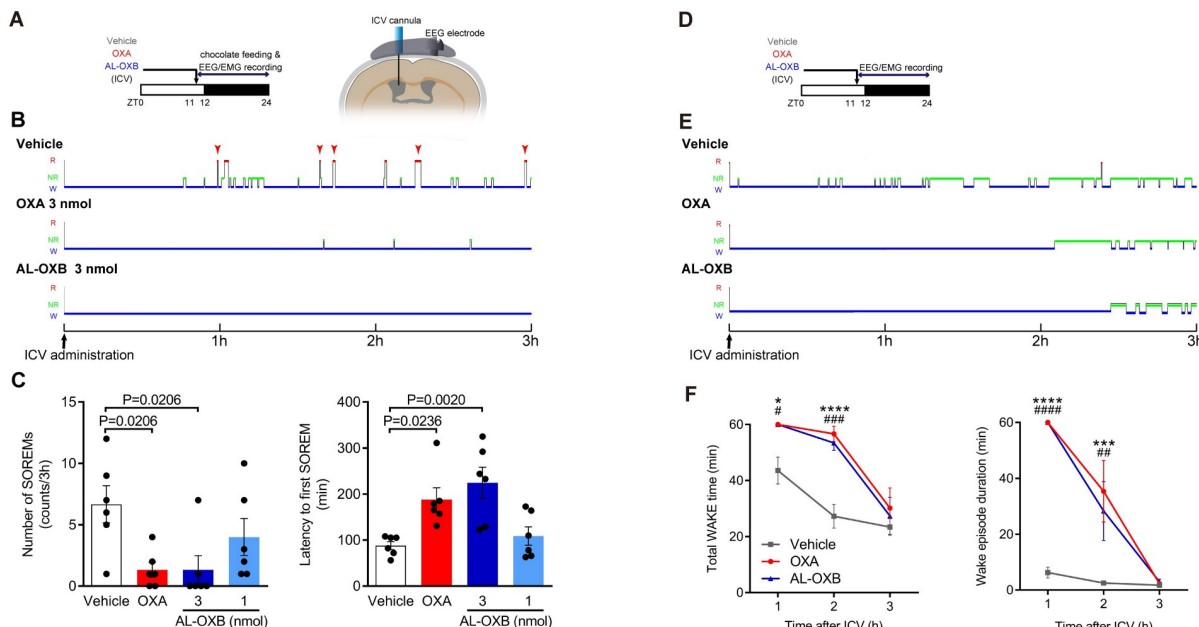

**Fig 2. ICV AL-OXB prevents both cataplexy-like episodes and fragmentation of wakefulness in OXKO mice. (A)** Experimental schedule (left) and surgery design (right). **(B)** Representative hypnograms showing the effect of ICV OXA or AL-OXB on cataplexy-like episodes in OXKO mice. R, REM state; NR, non-REM state; W, wake state. Red arrows: cataplexy-like states. **(C)** The number of cataplexy-like states during 3h (left) and latency to first cataplexy-like state (right) after ICV administration. Data represent the means ± SEM from 6 mice. Statistical analysis: one-way ANOVA followed by Bonferroni's multiple comparisons test. **(D)** Experimental schedule for evaluation of wakefulness fragmentation. **(E)** Representative hypnograms showing the effect of ICV OXA or AL-OXB on wakefulness fragmentation in OXKO mice. **(F)** Hourly plots of wake time (left) and wake episode duration (right) during 3 h after ICV administration. Data represent the means ± SEM from 6 mice. *p < 0.05, **p < 0.01, ****p < 0.0001 for OXA vs. vehicle; #p < 0.05, ##p < 0.01, ####p < 0.0001 for AL-OXB vs. vehicle; two-way repeated-measures ANOVA followed by Bonferroni's multiple comparisons test.

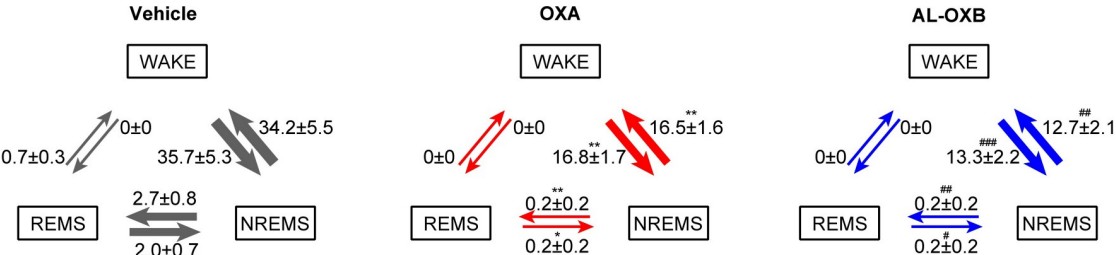

**Fig 3. ICV AL-OXB prevents sleep/wake fragmentation in OXKO mice.** The number of transitions between sleep/wake stages over 3 h after ICV administration of vehicle (left), 3 nmol OXA (middle), and 3 nmol AL-OXB (right) at ZT11-12. Data represent the means ± SEM from 6 mice. *p < 0.05, **p < 0.01, ****p < 0.0001 for OXA vs. vehicle; #p < 0.05, ##p < 0.01, ####p < 0.0001 for AL-OXB vs. vehicle; one-way ANOVA followed by Bonferroni's multiple comparisons test.

at 3 nmol also significantly decreased the number of cataplexy-like states and increased the latency to the first cataplexy-like state, but not at 1 nmol (Fig 2B and 2C) (we did not test OXA at 1 nmol). The amount of chocolate consumed after ICV administration did not differ significantly (vehicle: 2.0 ± 0.2 g; 3 nmol OXA: 2.3 ± 0.7 g; 3 nmol AL-OXB: 3.2 ± 0.8 g; 1 nmol AL-OXB: 2.5 ± 0.6 g).

We then examined whether selective activation of OX2R by AL-OXB is sufficient to ameliorate the fragmentation of wakefulness (defined here as the frequent transition between wakefulness and NREM sleep) during the dark phase in OXKO mice [19, 22] (Fig 2D). OXA (3 nmol) and AL-OXB (3 nmol) increased the total wake time and wake episode duration for 3 h after ICV administration (Fig 2E, 2F; S1 Fig). Vehicle injection also tended to increase the total wake time due to artifacts from ICV administration procedures. Accordingly, total NREM sleep time was similarly decreased by the injection of OXA or AL-OXB (S1C Fig). REM sleep was also suppressed, but did not reach statistical significance because there are few, if any, REM sleep episodes in the first few hours of injection (S1D Fig). Further, the number of transitions between wake and NREM was similarly reduced after the administration of OXA or AL-OXB (Fig 3). Taken together, our results suggest that a single ICV administration of AL-OXB, as well as OXA, ameliorates both cataplexy and fragmentation of wakefulness in OXKO mice. In other words, the selective activation of OX2R is sufficient to ameliorate both narcoleptic symptoms; a concomitant OX1R activation is not necessary for the therapeutic effects.

We next examined whether a long-term activation of OX2R can continue to suppress narcoleptic symptoms without acute desensitization. A 12-h continuous ICV administration of AL-OXB was given to OXKO mice during the dark phase. EEG/EMG signals were recorded on the day before, during the administration, and on the subsequent day (Fig 4A). OXA and AL-OXB both increased the total wake time and wake episode duration for 12 h during the continuous ICV administration in the dark phase (Fig 4B–4D, S2 Fig). Thus, OXA and AL-OXB maintained the efficacy over 12 h. Total wake time and wake episode duration in the subsequent dark phase after the administration showed a tendency to decrease compared with baseline (although not statistically significant), possibly reflecting a compensatory sleep rebound. There was no appreciable change in the NREM EEG delta power during the same period. We did not assess cataplexy-like states in this experiment, because the stress of continuous ICV infusion strongly suppressed the occurrence of cataplexy [32].

## Concomitant OX1R activation induces conditioned place preference in mice

Orexins play a role in the reward system [28, 33, 34]. We, therefore, examined whether AL-OXB-induced selective activation of OX2R induces addiction-related reinforcing effects in

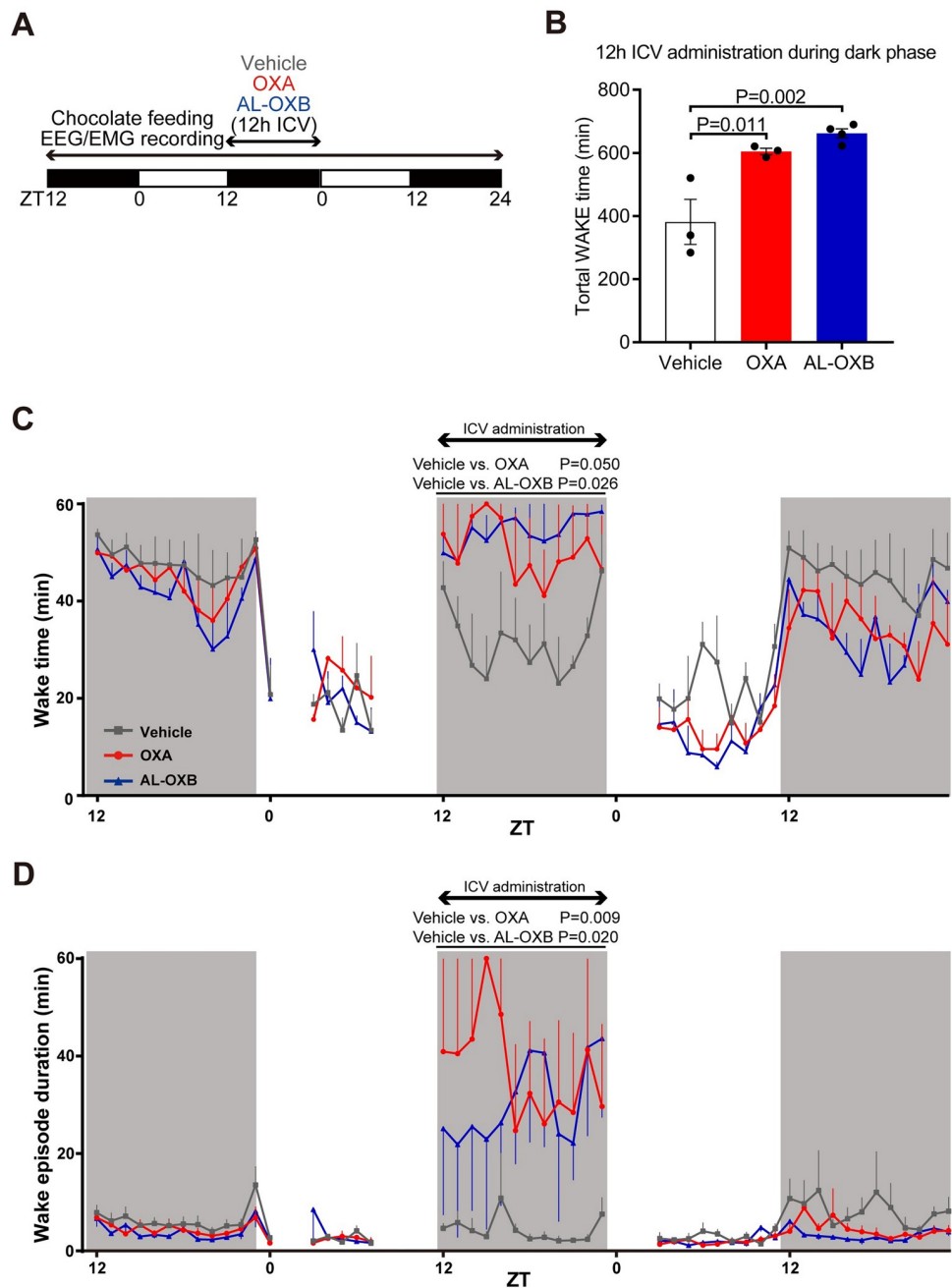

**Fig 4. Effect of continuous ICV administration of OXA and AL-OXB during dark phase on sleep-wake cycle in OXKO mice. (A)** Experimental schedule of ICV infusion and EEG/EMG recording. **(B)** Total wake time duration during 12-h ICV administration. Data represent the means ± SEM from 3 mice. Statistical analysis: one-way ANOVA followed by Bonferroni's multiple comparisons test. **(C and D)** Hourly plots of wake time (C) and wake episode duration (D) before, during and after continuous ICV administration. Data represent the means ± SEM from 3 mice. Statistical analysis: two-way repeated-measures ANOVA.

the conditioned place preference test conducted at ZT4-7 (Fig 5A). We chose to perform the conditioned place preference test during the light phase (resting phase for mice), since the endogenous orexin tones are lower in the light phase and therefore mice are more responsive to exogenous orexin. This allowed a more sensitive detection of the possible reinforcing effects

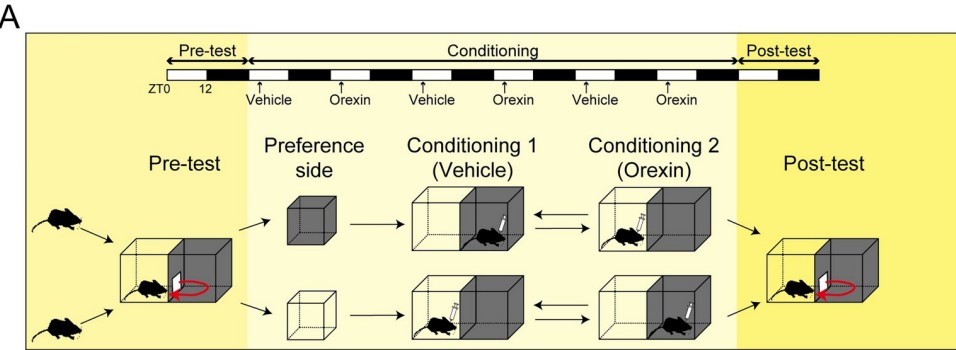

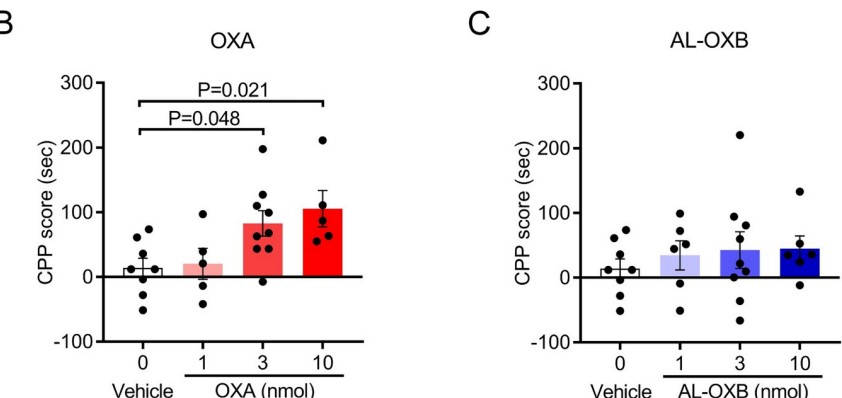

**Fig 5. OXA but not AL-OXB induces conditioned place preference. (A)** Experimental schedule of CPP test. **(B and C)** Effects of different doses of OXA (B) and AL-OXB (C) on CPP score. Same CPP data for vehicle injections are displayed twice in (B) and (C). Data represent the means ± SEM from 5 mice for 1 and 10 nmol OXA, 6 mice for 1 and 10 nmol AL-OXB, 9 mice for 3 nmol OXA or AL-OXB and 8 mice for vehicle. Statistical analysis: one-way ANOVA followed by Bonferroni's multiple comparisons test.

of ICV orexin. ICV administration of OXA (3 and 10 nmol) significantly produced conditioned place preference in wild-type mice (Fig 5B), whereas AL-OXB (1–10 nmol) did not (Fig 5C). Mice never slept during the 15-min pre-test or post-test periods under any of the drug conditions. These results suggest that the administration of an OX2R-selective agonist in sufficient doses to ameliorate cataplexy and fragmentation of wakefulness does not produce addiction-related behavior.

## Discussion

The vast majority (>90%) of type-1 narcoleptic patients are deficient in endogenous orexin peptides [35], which are non-selective OX1R/OX2R agonists. Therefore, in order to replace the orexin functions in the brain, it may require the activation of both OX1R and OX2R. However, it has been shown that OX2R is the main player in the orexin-mediated sleep/wake regulation. Indeed, OX2R-deficient mice, but not OX1R-deficient mice, exhibit cataplexy-like episodes and sleep/wake fragmentation, albeit more mildly than OX1R/OX2R double-deficient mice [11, 12]. These reports indicate that OX2R plays a pivotal role, whereas OX1R only a supplementary role, in sleep regulation. The aim of this study was to examine whether selective activation of OX2R is sufficient to rescue cataplexy and wake fragmentation in the mouse model of narcolepsy, providing a proof-of-concept for the mechanism-based treatment of narcolepsy with selective OX2R agonists. We demonstrated that AL-OXB, a selective OX2R

agonist, can fully ameliorate two major narcoleptic symptoms, cataplexy and fragmentation of wakefulness, in OXKO mice. We further showed that AL-OXB does not induce conditioned place preference, an addiction-related behavior.

OXA-induced increase of wakefulness and decrease of NREM sleep are attenuated not only in OX2R-deficient mice but also in OX1R-deficient mice [10]. Furthermore, restoration of OX1R expression in LC-noradrenergic neurons ameliorates fragmentation of wakefulness in OX1R/OX2R-deficient mice [20]. These reports suggest that OX1R is also involved in the transition and consolidation of wakefulness. However, selective activation of OX2R induced by AL-OXB fully ameliorated fragmentation of wakefulness in narcoleptic mice in the present study. Importantly, the therapeutically effective dose of AL-OXB induced c-fos selectively in OX2R-expressing TMN-histaminergic neurons but not in OX1R-expressing LC-noradrenergic neurons. This suggests that sleep/wake fragmentations in Type-1 narcoleptic patients may be adequately ameliorated by selective OX2R activation. In the absence of OX2R signaling, activation of OX1R may compensate for the lack of orexinergic signaling in OX1R/OX2R double-deficient mice, attenuating the wake fragmentation [20]. However, the activation of OX1R may be non-essential when OX2R signaling is provided.

Although AL-OXB was 3.6-times more potent than OXA on OX2R in $Ca^{2+}$ transient assays in vitro (Fig 1A), both peptides were approximately equipotent in vivo (Figs 2 and 4). A possible reason for this discrepancy might be a higher stability of OXA in vivo; OXA peptide might be more protected from proteolysis due to the N-terminal pyroglutamate residue and the two intrachain disulfide bonds. We previously reported that the small-molecule selective OX2R agonist YNT-185 effectively suppressed cataplexy-like episodes in OXKO mice, but the compound did not appreciably ameliorate the wake fragmentation [19]. In contrast, here we demonstrated that AL-OXB can ameliorate both narcoleptic symptoms in OXKO mice. The difference in the potency of these agonists may account for the discrepancy: the in vitro $EC_{50}$ value of AL-OXB for OX2R is approximately 200-fold lower than that of YNT-185. Although more research will be required, it is likely that higher agonist potency may be required for ameliorating wake fragmentation than suppressing cataplexy. Indeed, a previous study with inducible genetic ablations of orexin neurons in mice indicated that, whereas the ablation of 80% of orexin neurons results in wake fragmentation, a 95% ablation is required to induce cataplectic phenotype [36]. Pharmacologic partial blockade of OX1R/OX2R in mice and humans can readily induce wake fragmentation and somnolence, while it is rather difficult to induce cataplexy by orexin receptor antagonism [32]. These findings are consistent with the idea that the level of orexin tone required to prevent wake fragmentation is higher than that required to inhibit cataplexy.

Orexin also activates the reward system [13, 37]. Orexin neurons in the LH project to the dopaminergic reward system in the ventral tegmental area (VTA) [38], and promote the dopamine release in the nucleus accumbens and the prefrontal cortex [28, 33, 39], potentially inducing addiction-like behaviors and reinforcing effects. Orexin-mediated neural signaling also contributes to the development of addiction to drugs of abuse such as cocaine, morphine and nicotine [40–42]. Anatomically, both OX1R and OX2R are present in VTA, with dopaminergic neurons predominantly expressing OX1R [18]. Moreover, addiction-like behaviors induced by drugs of abuse or OXA itself (non-selective OX1R/OX2R agonist) [33, 34] are inhibited by OX1R-selective antagonist SB334867 [43–45], suggesting that OX1R-mediated signaling may play an important role in addiction-like behaviors. On the other hand, it has also been reported that morphine-induced rewarding effects are attenuated by OX2R antagonists [43, 44, 46]. Therefore, it was essential to examine whether selective activation of OX2R can induce addiction-like behaviors. In the present study, OXA significantly induced conditioned place preference in mice, whereas AL-OXB did not. This suggests that the activation of

OX1R is required for the induction of addiction-like behaviors. Thus, selective OX2R activation may provide sufficient therapeutic effects for narcoleptic symptoms in mice, without promoting addiction-like behaviors or reinforcing effects.

Limitations in the present study include the use of the peptidic agonists, which necessitated central administration by ICV injections. Here we prioritized the importance of a side-by-side comparison of high-potency/efficacy agonists with different receptor subtype selectivity over the practical applicability of peripherally administrable orexin agonists. To our knowledge, there is no non-peptidic, high-affinity OX1R or OX1R/OX2R agonists available yet. In addition, further studies are necessary to mechanistically dissect the roles of OX1R- versus OX2R-mediated neuronal circuits in the regulation of sleep/wake and addictive behaviors. Also, in this study we used OXKO mice as a model of narcolepsy. In contrast, human narcolepsy type-1 is caused by autoimmune destruction of orexin neurons, which contain additional neurotransmitters such as glutamate, dynorphin and neuronal activity-regulated pentraxin (Narp) [47–50]. Thus, orexin neuron-ablated mouse models [36, 51] may more closely mimic the pathology of human narcolepsy type-1. However, to our knowledge, there is no report positively documenting a sleep/wake-related role for these non-orexin transmitters colocalized in orexin neurons; all sleep/wake abnormalities of orexin neuron-ablated mice have thus far been attributed to the deficiency of orexin peptides. Finally, the number of mice in the chronic ICV infusion experiments (Fig 4) are very small (n = 3); as such, data should be interpreted with caution and considered mainly descriptive rather than quantitative.

In conclusion, we reported here that OX2R-selective agonism is sufficient to ameliorate both cataplexy (related to REM sleep gating) and fragmentation of wakefulness (NREM gating) in narcoleptic mice without inducing addiction-related behaviors. These findings support the notion that selective OX2R agonists may be an effective and safe therapeutic strategy for the mechanism-based treatment of narcolepsy.

## Supporting information

**S1 Fig. Effect of ICV administration of OXA and AL-OXB on the fragmentation of wakefulness in OXKO mice (related to Fig 2 of the main text). (A-D)** Hourly plots of wake time (A), wake episode duration (B), NREM sleep time (C) and REM sleep time (D) during dark phase after ICV administration. **(E and F)** First 3 hours of the same EEG/EMG data analyzed in 4-s epochs; wake time (E) and wake episode duration (F). Data represent the means ± SEM from 6 mice. *$p < 0.05$, **$p < 0.01$, ****$p < 0.0001$ for OXA vs. vehicle; #$p < 0.05$, ##$p < 0.01$, ####$p < 0.0001$ for AL-OXB vs. vehicle; two-way repeated-measures ANOVA followed by Bonferroni's multiple comparisons test.
(PDF)

**S2 Fig. Effect of continuous ICV administration of OXA and AL-OXB during dark phase on sleep-wake cycle in OXKO mice (related to Fig 4 of the main text). (A)** Experimental schedule of ICV infusion and EEG/EMG recording. **(B and C)** Hourly plots of NREM sleep time (B) and REM sleep time (C) before, during and after continuous ICV administration. Data represent the means ± SEM from 3 mice. Statistical analysis: two-way repeated-measures ANOVA.
(PDF)

**S1 Data. Raw data for Figs 1–5 and S1 and S2 Figs.**
(XLSX)

## Acknowledgments

We thank all Yanagisawa/Funato laboratory members and International Institute for Integrative Sleep Medicine members for their support, technical assistance and discussion.

## Author Contributions

**Conceptualization:** Masashi Yanagisawa.

**Formal analysis:** Hikari Yamamoto.

**Funding acquisition:** Masashi Yanagisawa.

**Investigation:** Hikari Yamamoto, Yasuyuki Nagumo, Yukiko Ishikawa, Yoko Irukayama-Tomobe, Yukiko Namekawa, Tsuyoshi Nemoto, Hiromu Tanaka, Genki Takahashi, Akihisa Tokuda.

**Resources:** Tsuyoshi Saitoh.

**Supervision:** Hiroshi Nagase, Hiromasa Funato, Masashi Yanagisawa.

**Writing – original draft:** Hikari Yamamoto.

**Writing – review & editing:** Yasuyuki Nagumo, Masashi Yanagisawa.

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
