## [Decision Letter · Decision Letter 0]

4 Apr 2022

PONE-D-22-06089OX2R-selective orexin agonism is sufficient to ameliorate cataplexy and sleep/wake fragmentation without inducing drug-seeking behavior in mouse model of narcolepsyPLOS ONE

Dear Dr. Yanagisawa,

Thank you for submitting your manuscript to PLOS ONE. Your manuscript was evaluated by four experts in the field and by myself. After careful consideration, we all feel that it has merit but does not fully meet PLOS ONE’s publication criteria as it currently stands, with particular reference to the criteria that experiments, statistics, and other analyses are performed to a high technical standard and are described in sufficient detail and that conclusions are presented in an appropriate fashion and are supported by the data. Therefore, we invite you to submit a revised version of the manuscript that addresses the points raised during the review process. In addition to the points raised by all Reviewers, please also address my additional editor comments below.

We look forward to receiving your revised manuscript.

Kind regards,

Alessandro Silvani, M.D., Ph.D.

Academic Editor

PLOS ONE

Journal Requirements:

[NO authors have competing interests]. 

Additional Editor Comments:

For clarity and ease of consultation, please report in the manuscript the administered concentrations of AL-OXB and OXA in terms of net peptide. Please indicate the purity and % net peptide content for each peptide. 

Please clarify whether the data fulfilled the requirements for parametric statistical analysis with ANOVA.

Please note that a working URL address to all data underlying the findings or inclusion of the data as supplementary material will be needed before the eventual acceptance of the manuscript for publication.

Reviewers' comments:

Reviewer's Responses to Questions

**Comments to the Author**

1. Is the manuscript technically sound, and do the data support the conclusions?

Reviewer #1: Yes

Reviewer #2: Partly

Reviewer #3: Yes

Reviewer #4: Partly

2. Has the statistical analysis been performed appropriately and rigorously? 

Reviewer #1: Yes

Reviewer #2: Yes

Reviewer #3: Yes

Reviewer #4: Yes

3. Have the authors made all data underlying the findings in their manuscript fully available?

Reviewer #1: Yes

Reviewer #2: No

Reviewer #3: No

Reviewer #4: No

4. Is the manuscript presented in an intelligible fashion and written in standard English?

Reviewer #1: Yes

Reviewer #2: Yes

Reviewer #3: Yes

Reviewer #4: Yes

5. Review Comments to the Author

Reviewer #1: This is a very nicely done paper describing the possible roles of the two orexin receptors in reversing the symptoms of narcolepsy. The combination of in vitro and in vivo studies is a great strength of the study, as is the conditioned place preference tests. The paper shows the way to a possible treatment of human narcolepsy.

A limitation of the study is the use of orexin KO mice. Although these mice, developed by some of these investigators, led rather directly to the cause of human narcolepsy, the consensus in the field is that human narcolepsy is due to the loss of orexin neurons, not just the orexin peptide. Orexin neurons contain glutamate, dynorphin and Narp. These transmitters are present in the KO mice, but not in human narcoleptics, so the KO mouse results may not exactly align with the pathology of human narcolepsy. The use of the orexin neuron deletion mice developed by Yamanaka might more closely parallel the human condition and comparison of the results of these two models might shed light on the role of the other transmitters in orexin neurons.

In the introduction, I would suggest moving or deleting "and dreaming." Human cataplexy is generally not accompanied by dreaming unless it progresses into a long REM sleep episode. You may want to position these words after "hallucinations." Histaminergics is misspelled.

A very minor point is that the support and animal study questions ask by PLOS One are not properly addressed in their forms, although all the information is in the manuscript.

Reviewer #2: The authors examined whether orexin 2 receptor selective agonist (AL-OXB) could effectively reduce cataplexy and sleep/wake fragmentation in orexin knockout mice and got affirmative results. In addition, using wild type mice, the authors showed that ICV administration of OXA but not AL-OXB induced conditioned preference indicating less addiction-inducing effect in the latter. The theme of the study is worth examining and overall conclusion seems convincing. However, there are several points to be clarified.

1) Please report the amount of chocolate consumed in the cataplexy experiment (Fig. 2) because orexin is known to induce food seeking. Was there any possibility that possible change in appetite affected the wake/sleep structure?

2) Please specify when of the day pre- and post-test was performed in CPP testing. According to Fig. 5A, it appeared to be during the daytime around ZT3. If it is correct, then mice should be in the resting period and sometimes fall into asleep. Such behavior may distort the preference score. To exclude such possibility, information should be provided about mobility such as the crossing times between the components and/or distance traveled during the observation period.

3) In related to the point #2, why the authors selected to test and inject drugs during the resting period whereas the drugs were injected just before the active (night) period in the sleep study? To examine possible side effect (addiction-inducing effect in this case) of a drug, we should use the same dosing schedule in the main effect (sleep study in this case). Don’t you agree?

Reviewer #3: The manuscript aims at demonstrating that the selective activation of the orexin receptor 2 is sufficient to ameliorate cataplexy and sleep fragmentation in orexin-deficient (ORX-KO) mice without inducing drug seeking behavior as tested using the place preference test.

The originality of this study is to directly compare OXA and AL-OXB in vivo. The authors show that only TMN neurons are activated by AL-OXB (expressing only ORX receptor 2) while OXA activates both, neurons of the TMN and the LC (expressing only ORX receptor 1) as a validation of the selectivity of the compounds for the orexin receptor 2.

Although the study has limits, it is of interest for the design of future therapeutics.

The manuscript is well written.

Comments

• Please add the total number of mice used in the study and for each experiments in the method section.

• Chronic icv injections were performed in only 3 mice. No statistical test can be robust with such small number of individuals. Data should be reported with caution and be mainly descriptive (rather than quantitative). Please mention/discuss it in a “technical limitation” section.

• Figure 1 shows the dose response curves of intracellular CA2+ induced by OXA and L-OXB in vitro. L-OXB is approximately one log more efficient than OXA,. However, drugs are used at the same concentration in vivo. Could the authors comment on it? Please discuss it.

• Figure S1 show an increase in NREM sleep after OXA treatment compared to vehicle at ZT5-7, that is not seen with L-OXB. It might be an additional positive point for L-OXB versus ORXA, but it is not discussed. Could the authors comment on it?

Reviewer #4: This important paper evaluates the efficacy of orexin A (OXA) and AL-orexin B (AL-OXB) in several in vitro and in vivo assays and concludes that Ox2R activation by AL-OXB is sufficient to prevent cataplexy-like episodes and fragmentation of wakefulness in OXKO mice. The work is well-thought out, carefully executed and, with one exception, the results are properly interpreted. That exception is particularly evident on page 17 when the authors conclude the first paragraph with the statement “In other words, the selective activation of OX2R is sufficient to ameliorate both narcoleptic symptoms.” I understand that the authors would like to make this conclusion but, based on the data presented in Figs. 2C, 2D and Fig. 4, the same statement could be made for OX1R since OXA is at least as effective as AL-OXB in all the sleep/wake bioassays shown. The data presented does not exclude this possibility because an Ox1R-selective compound has not been evaluated. As such, I find that statement as well as the title of the paper to be misleading.

The EEG/EMG recordings are scored in 20-sec epochs which is unusually long for rodent studies in which 10-sec epochs are the norm and many labs even score in 4-sec epochs. Although the choice of epoch duration is somewhat arbitrary, longer epochs provide less resolution of the fine structure of sleep architecture. In fact, it is certainly conceivable that all 3 states could occur in a 20-sec epoch. Consider, for example, the sequence NREM-REM-Wake. Consequently, the authors should describe their rationale for such long epochs.

Related to this issue are the determinations of “mean episode duration” as shown in Figs. 2F, 4D and Fig. S1. In fact, Fig. 3 is entirely leveraged on episode duration measurements as the number of state transitions reported in this figure would certainly be larger if 10-sec epochs were used. Because of the fact that rodent sleep is fragmented throughout the 24-h period rather than consolidated as occurs in (most) humans, many sleep researchers establish rules to determine a bout (or episode) of NREM or REM sleep, such as that 3 consecutive epochs of NREM sleep or 2 consecutive epochs of REM sleep must occur before a bout (or episode) is consider to be NREM or REM sleep. The authors do not stipulate any such criteria, which may be due to their use of the long epochs.

Neither OXA and AL-OXB at 1 nmol induce CPP but there’s no data to show whether OXA at 1 nmol could prevent cataplexy-like episodes and fragmentation of wakefulness in OXKO mice. Has OXA been evaluated at that dose or not?

What effect does continuous ICV infusion as shown in Fig. 4 have on cataplexy, if any? It would be interesting to know what effect, if any, continuous ICV infusion of OXA and AL-OXB have on body temperature.

Supp. Fig. 2 suggests that there is a trend toward a rebound in NREM sleep on both the day and subsequent night after cessation of ICV infusion. Although these trends may not be significant on an hourly basis, are they significant over 6-h or 12-h periods? What about NREM delta power?

On page 17, the authors conclude the next paragraph with the statement “despite the stress.” What stress do the authors refer to? This is the first use of the word “stress” in the paper.

On page 20, the authors state “it is likely that higher agonist potency may be required for ameliorating wake fragmentation than suppressing cataplexy.” This is an interesting point that should be developed further.

Figure 4: why are the error bars so large during the infusion? Figure 4D suggests that AL-OXB is weaker than OXA.

Specific comments

p. 10: “administrated” should be “administered”.

P. 17, last sentence would be better English if the authors changed it to read “These results suggest that the administration of an OX2R-selective agonist in sufficient doses to ameliorate cataplexy and fragmentation of wakefulness does not produce addiction-related behavior.”

p. 21, first line: eliminate “the” after “in”.

The authors are not appropriately listed in Ref 25.

6. PLOS authors have the option to publish the peer review history of their article (what does this mean?). If published, this will include your full peer review and any attached files.

Reviewer #1: **Yes: **Jerome Siegel

Reviewer #2: No

Reviewer #3: No

Reviewer #4: No

---

## [Author Response · Author response to Decision Letter 0]

18 May 2022

Our detailed point-by-point responses to specific comments from editor and reviewers are listed in the separate "Response to Reviewers" file.

---

## [Decision Letter · Decision Letter 1]

5 Jun 2022

PONE-D-22-06089R1OX2R-selective orexin agonism is sufficient to ameliorate cataplexy and sleep/wake fragmentation without inducing drug-seeking behavior in mouse model of narcolepsyPLOS ONE

Dear Dr. Yanagisawa,

Thank you for submitting your manuscript to PLOS ONE. After careful consideration, we feel that it has merit but does not fully meet PLOS ONE’s publication criteria as it currently stands. Therefore, we invite you to submit a revised version of the manuscript that addresses the points raised during the review process. In particular, I encourage you to make minor revisions to your manuscript in order to address the final comments by Reviewer 1 and Reviewer 4. 

We look forward to receiving your revised manuscript.

Kind regards,

Alessandro Silvani, M.D., Ph.D.

Academic Editor

PLOS ONE

Journal Requirements:

Reviewers' comments:

Reviewer's Responses to Questions

**Comments to the Author**

1. If the authors have adequately addressed your comments raised in a previous round of review and you feel that this manuscript is now acceptable for publication, you may indicate that here to bypass the “Comments to the Author” section, enter your conflict of interest statement in the “Confidential to Editor” section, and submit your "Accept" recommendation.

Reviewer #1: All comments have been addressed

Reviewer #2: All comments have been addressed

Reviewer #3: All comments have been addressed

Reviewer #4: (No Response)

2. Is the manuscript technically sound, and do the data support the conclusions?

Reviewer #1: Yes

Reviewer #2: (No Response)

Reviewer #3: Yes

Reviewer #4: Yes

3. Has the statistical analysis been performed appropriately and rigorously? 

Reviewer #1: Yes

Reviewer #2: (No Response)

Reviewer #3: Yes

Reviewer #4: Yes

4. Have the authors made all data underlying the findings in their manuscript fully available?

Reviewer #1: Yes

Reviewer #2: (No Response)

Reviewer #3: Yes

Reviewer #4: Yes

5. Is the manuscript presented in an intelligible fashion and written in standard English?

Reviewer #1: Yes

Reviewer #2: (No Response)

Reviewer #3: Yes

Reviewer #4: Yes

6. Review Comments to the Author

Reviewer #1: I don't understand what the authors are saying in the second sentence of the Introduction. I suggest shortening it to: "The abnormal gating of REM sleep-related neurophysiological mechanisms, such as REM atonia contributes to cataplexy (sudden bilateral skeletal muscle weakening triggered by a strong emotion) and sleep paralysis."

Dreaming is not confined to REM sleep and is not studied (nor could it be) in this mouse experiment.

Reviewer #2: (No Response)

Reviewer #3: All my comments were answered, I am satisfied by the way they were taken in considerations

Thank you

Reviewer #4: The authors' response re the use of the 20-sec epoch is very misleading and incorrect. For example, in a 20-sec epoch, the following sequence of states commonly occurs: NREM-REM-W. This epoch would be scored as one of the 3 states. The purpose of contiguity rules for shorter 4-sec or 10-sec epochs is to obviate this problem when sleep architecture analyses are reported (e.g., number of bouts and mean bout duration per state). The authors' revised text is thus both facile and misleading and must be corrected before I recommend acceptance.

7. PLOS authors have the option to publish the peer review history of their article (what does this mean?). If published, this will include your full peer review and any attached files.

Reviewer #1: No

Reviewer #2: **Yes: **Tomoyuki Kuwaki

Reviewer #3: No

Reviewer #4: No

---

## [Author Response · Author response to Decision Letter 1]

7 Jul 2022

Our point-by-point responses to Reviewers' comments are fully described in the "Response to Reviewers" file.

---

## [Decision Letter · Decision Letter 2]

11 Jul 2022

OX2R-selective orexin agonism is sufficient to ameliorate cataplexy and sleep/wake fragmentation without inducing drug-seeking behavior in mouse model of narcolepsy

PONE-D-22-06089R2

Dear Dr. Yanagisawa,

We’re pleased to inform you that your manuscript has been judged scientifically suitable for publication and will be formally accepted for publication once it meets all outstanding technical requirements.

Kind regards,

Alessandro Silvani, M.D., Ph.D.

Academic Editor

PLOS ONE

Additional Editor Comments (optional):

Reviewers' comments:

Reviewer's Responses to Questions

**Comments to the Author**

1. If the authors have adequately addressed your comments raised in a previous round of review and you feel that this manuscript is now acceptable for publication, you may indicate that here to bypass the “Comments to the Author” section, enter your conflict of interest statement in the “Confidential to Editor” section, and submit your "Accept" recommendation.

Reviewer #4: All comments have been addressed

2. Is the manuscript technically sound, and do the data support the conclusions?

Reviewer #4: (No Response)

3. Has the statistical analysis been performed appropriately and rigorously? 

Reviewer #4: (No Response)

4. Have the authors made all data underlying the findings in their manuscript fully available?

Reviewer #4: (No Response)

5. Is the manuscript presented in an intelligible fashion and written in standard English?

Reviewer #4: (No Response)

6. Review Comments to the Author

Reviewer #4: (No Response)

7. PLOS authors have the option to publish the peer review history of their article (what does this mean?). If published, this will include your full peer review and any attached files.

Reviewer #4: No

---

## [Editor Report · Acceptance letter]

14 Jul 2022

PONE-D-22-06089R2 

OX2R-selective orexin agonism is sufficient to ameliorate cataplexy and sleep/wake fragmentation without inducing drug-seeking behavior in mouse model of narcolepsy 

Dear Dr. Yanagisawa:

I'm pleased to inform you that your manuscript has been deemed suitable for publication in PLOS ONE. Congratulations! Your manuscript is now with our production department. 

Kind regards, 

on behalf of

Prof. Alessandro Silvani 

Academic Editor

PLOS ONE